# Directionally-Dependent Mechanical Properties of Ti6Al4V Manufactured by Electron Beam Melting (EBM) and Selective Laser Melting (SLM)

**DOI:** 10.3390/ma14133603

**Published:** 2021-06-28

**Authors:** Tim Pasang, Benny Tavlovich, Omri Yannay, Ben Jackson, Mike Fry, Yuan Tao, Celine Turangi, Jia-Chang Wang, Cho-Pei Jiang, Yuji Sato, Masahiro Tsukamoto, Wojciech Z. Misiolek

**Affiliations:** 1Department of Manufacturing and Mechanical Engineering and Technology, Oregon Institute of Technology, Klamath Falls, OR 97601, USA; 2Metal Technology Center, Rafael Ltd., Haifa 3102102, Israel; bennyta@rafael.co.il (B.T.); omri.yannay@rafael.co.il (O.Y.); 3TiDA Ltd., Tauranga 3116, New Zealand; ben@tida.co.nz (B.J.); mike.fry@tida.co.nz (M.F.); 4Department of Mechanical Engineering, Auckland University of Technology, Auckland 1020, New Zealand; yuan.tao@aut.ac.nz (Y.T.); celine.turangi@aut.ac.nz (C.T.); 5Department of Mechanical Engineering, National Taipei University of Technology, Taipei 10608, Taiwan; jcw@mail.ntut.edu.tw (J.-C.W.); jcp@mail.ntut.edu.tw (C.-P.J.); 6Joining and Welding Research Institute (JWRI), Osaka University, Osaka 565-0871, Japan; sato@jwri.osaka-u.ac.jp (Y.S.); tukamoto@jwri.osaka-u.ac.jp (M.T.); 7Loewy Institute, Department of Materials Science and Engineering, Lehigh University, Bethlehem, PA18015, USA; wzm2@lehigh.edu

**Keywords:** additive manufacturing, electron beam melting, selective laser melting, wrought alloy

## Abstract

An investigation of mechanical properties of Ti6Al4V produced by additive manufacturing (AM) in the as-printed condition have been conducted and compared with wrought alloys. The AM samples were built by Selective Laser Melting (SLM) and Electron Beam Melting (EBM) in 0°, 45° and 90°—relative to horizontal direction. Similarly, the wrought samples were also cut and tested in the same directions relative to the plate rolling direction. The microstructures of the samples were significantly different on all samples. α′ martensite was observed on the SLM, acicular α on EBM and combination of both on the wrought alloy. EBM samples had higher surface roughness (Ra) compared with both SLM and wrought alloy. SLM samples were comparatively harder than wrought alloy and EBM. Tensile strength of the wrought alloy was higher in all directions except for 45°, where SLM samples showed higher strength than both EBM and wrought alloy on that direction. The ductility of the wrought alloy was consistently higher than both SLM and EBM indicated by clear necking feature on the wrought alloy samples. Dimples were observed on all fracture surfaces.

## 1. Introduction

Additive manufacturing (AM) has gained huge popularity in the last decade or so. Its ability to produce complex shapes, reduce the number of parts, reduce the material waste, etc. have been the main drive of this technology. According to ASTM, AM can be devided into seven categories including (i) VAT photopolymerisation, (ii) material jetting, (iii) binder jetting, (iv) sheet lamination, (v) material extrusion, (vi) powder bed fusion (PBF), and (vii) directed energy deposition (DED). For metallic materials, Directed Energy Deposition (DED) and Powder Bed Fusion (PBF) are the only options. Within the PBF methods, electron beam melting (EBM) and selective laser melting (SLM) are two common methods to produce metallic parts where the powder is melted and fused by either electron beam or laser beam. SLM materal is generally stronger and less ductile compared with EBM [1]. SLM produces martensite due to ambient temperature manufacturing bed with fast cooling, while EBM produces alpha lath (alpha-beta) structure due to the powder bed being maintained at much higher temperature for manufacturing (approximately 700 °C for Ti6Al4V). Manufacturing in SLM is typically carried out under inert gas, while EBM is performed in vacuum athmosphere with powder bed heated to above stress relieving temperature, hence, minimising the residual stresses caused during solidification [1,2]. Ductility of SLM-ed parts can be improved by HIP or heat treatment. The details of the SLM and EBM processes have been described by various authors [3,4].

As far as the metallic material is concerned, some of the metals/alloys which are readily used for AM including stainless steels (SS316 and SS304), nickel-based alloys (Inconel 718/625), aluminium alloys (AlSi10Mg, Scalmalloy), Co-Cr alloy, tool steels (H13), and titanium (Ti6Al4V and CP Ti). The above metallic AM has been reviewed by a few authors [5,6,7,8,9].

More specifically, evaluation of mechanical properties of AM of Ti6Al4V produced by EBM and SLM have been reported separately by various authors [10,11,12,13,14,15,16,17,18,19,20,21]. Limited studies have been conducted to compare SLM and EBM [22,23,24,25,26,27,28], and there has been even more limited work to directly compare the AM materials with wrought alloys. The reported results varied. In general, however, SLM produces samples with higher strength (lower ductility), higher fatigue strength compared to EBM products due to the faster cooling rates on the former [24,25,26,27,28]. Gao et.al. specifically reported that the higher yield and tensile strengths of samples produced by SLM compared with those of EBM due to the presence of α′ martensite on the former. They also reported SLM and EBM samples show comparable ductility, fatigue strength and hardness [26].

EBM and SLM are the two most common methods to produce metallic componets, therefore, this investigation was dedicated to focus on the Ti6Al4V samples manufactured by EBM and SLM, and compared to the traditional wrought alloys. To the best knowledge of the authors, there has been no publications on a comprehensive comparison between EBM, SLM and wrought alloys comparing various build directions, hence, the novelty of this paper. It presents a comparison of surface roughness, microsctructures and mechanical properties of samples manufactured by EBM and SLM built in the 0°, 45° and 90° in the as-printed conditions, compared with those of wrought alloys (annealed and solution treated and aged—STA conditions) also in three different directions. 

## 2. Experimental Procedures

### 2.1. Materials and Sample Directions

The material investigated in this study was Ti-6Al-4V (Ti64) manufactured by three different processes, i.e., (i) wrought and rolled sheet, hereafter wrought (ii) electron beam melting-EBM, and (iii) selective laser melting-SLM with a thickness of around 2 mm (Figure 1). For the wrought alloy, the samples were cut in 0° (longitudinal/horizontal), 45° and 90° (transverse), while the EBM and SLM samples were built also in the same directions (Figure 1). The particle size of SLM and EBM powders were 5–50 μm and 50–150 μm, respectively. The powder for EBM was provided by Arcam (AP&C) Co. (ARCAM, Mölndal Municipality, Gothenburg, Sweden), while the SLM powder was provided by Pyrogenesis Canada Inc. (Pyrogenesis, Montréal, QC, Canada).

The EBM samples were printed using Arcam Q10plus 5.0 themes with a layer thickness of 50 μm, speed function of 64, line offset of 0.2 mm and a current of 30 mA which would give an energy density of around 94 J/mm^3^. The SLM samples were printed using EOS GmbH, EOSINT M270 machine using a rotating stripe pattern. Parameters of 190 W laser power, scanning velocity of 1200 mm/s, hatch distance of 100 μm, and layer thickness of around 30 μm were used, resulting in a calculated energy density of 53 J/mm^3^.

The densities of the printed samples were measured by the Archimedes methods.

### 2.2. Surface Roughness

Surface roughness, Ra, was measured according to layer direction using Taylor Hobson Ultra 2006 version software. An average of three readings was taken per sample.

### 2.3. Metallography and Microscopy

Cross-sectional metallography samples were prepared from each material. The samples were ground were ground using SiC papers with grit size from 100 to 2400, and were polished up to 0.3 μm diamond paste. A diluted hydrogen peroxide (30%) was used at the final stage of polishing for a better response to ecthing. The samples were then etched with Kroll’s reagent (3 mL HF + 6 mL HNO_3_ + 100 mL water). An optical microcope (Olympus PME3, Tokyo, Japan), was employed to observed the microctructure. The microscope is equipped with an image analysis (ImageJ software, version 1.51, NIH, Bethesda, MD, USA). A scanning electron microscope (Hitachi SU-70 Schottky, Marunouchi, Chiyoda-ku, Tokyo, Japan) was employed to examine the surface morphology of the as-printed samples and the fracture surfaces of the tensile tested samples.

### 2.4. Mechanical Testing

Vicker’s hardness tests were performed on the metallographically-prepared samples using a load of 300 g and a dwell time of 10 s. A minimum of three hardness indentations were made for each sample.

Figure 2 shows the schematic drawings of a dog-bone sample for tensile testing. The samples built by SLM and EBM are shown in Figure 1 while the wrought samples, they were cut from a rolled sheet, also in three different directions as indicated on Section 2.1. Tensile tests were conducted on Shimadzu AG-X Plus 300KN load frame with pneumatic grips and with a speed of 3 mm/min. The machine was equipped with a Correlated Solution 3D DIC system with a couple of 2.3MP cameras. This non-contact technique was employed to measure both the overall strains as well as the local strains at the fracture locations. Prior to the real tensile testing, some trials were conducted at a speed of 0.15 mm/min as suggested by ASTM, but the time required to test a sample was significantly too long. In addition, there were no significant differences observed in terms of physical appearance and tensile data with those tested at 3 mm/min, therefore, all the samples were tested at a speed of 3 mm/min. The post processing of the DIC was also done by the Correlated Solution VIC3D.

## 3. Results and Discussion

### 3.1. Chemical Composition and Densities

The chemical composition of the samples is presented on Table 1.

The relative density of the samples were 96–98% for SLM, 96–97% EBM and >99% for the sheet wrought alloy. These are equal to the density of 4.25–4.34 g/cm^3^, assuming 100% is 4.432 g/cm^3^. Note that porosity is defined as the volume fraction of pores. It is expressed as p = 1 − ρ/ρ_T_, where ρ is the sample density and ρ_T_ is the theoretical density of the substance [29,30]. The theoretical density of Ti-6Al-4V was 4432 kg.m^−3^ or 4.432 g/cm^3^ as indicated above.

### 3.2. Surface Morphology and Hardness

Figure 3 shows the surface morphology of the printed samples. They looked similar to what have been reported previously [27,28]. However, with the reported AM samples, they did not show any indication of surface cracking as reported elsewhere [27]. It is noticeable that the particle size of SLM was smaller than those of EBM, i.e., 10–40 µm and 20–70 µm, respectively. The size of SLM particle was comparable to those used by other researchers, e.g., 5–50 µm [20] and 25–45 µm [28,29,30,31,32,33]. Other works with EBM indicated the particle size of around 82 µm [15], 77 µm [28], 60–70 µm [34,35]. According to Froes [1] the desired powder size of SLM is 20–75 µm, and 40–150 µm for EBM. The main reason why EBM uses larger particles is to keep them stick to the building platform instead of “flying around” (dusting effect) inside the chamber during the printing process [26]. Small size particles may cause particles spreading in the chamber due to electrostatic discharge [32]. From Figure 3, it is a also apparent that all surfaces contained partially melted particles which are attached to the surface on samples from both AM processes which may have contributed to the surface roughness.

Comparing the roughness of the AM samples in these experiments with other reports, they were smoother than those reported by others as described below. Figure 4 shows the surface roughness of the samples in this investigation was 6–8 μm, 18–23 μm and 0.45–0.6 μm for SLM, EBM and wrought samples, respectively. The relatively rougher surface on EBM samples is likely due to its larger particle size compared with that of SLM, particularly those of the partially melted particles which are still intact on the surface. This is consistent with what was reported by Zhao et al. [28]. A number of articles reported similar surface roughness for SLM, e.g., 30–40 [19,27]. Meanwhile, Song et al., using particle size of around 10–31 µm, claimed that with the correct parameters i.e., 110 W and a scanning speed of 0.4 m/s, a smooth surface of Ti6Al4V can be fabricated with SLM due to the continuous melting effect [33]. For EBM, surface roughness has been reported to be around 46 µm [18], 70 µm [27], and up to 130 µm [34]. Furthermore, Fousova et al. [27] indicated that the larger size of particles in EBM increased the unevennes of the surface. Froes [1] suggested that SLM parts may have surface roughness of about 10–25 µm, while EBM could be as high as 50 µm which is as rough as sand casting products. Apart from the material’s feedstock, i.e., size of particle and improper melting, the surface roughness may also be associated with post processing such as CNC machined, scan path strategy and powder morphology [6,36]. For certain applications such as implants, rough surfaces are sometimes preferred to facilitate bone ingrowth. 

The microstructure of SLM showed a mixture of α’ martensite and acicular α, while the EBM samples had acicular α and prior β grain boundaries (Figure 5). These findings are consistent with what has been reported previously [7,8,9,10,11,12,20,24,27,28]. This is due to the nature of the processes, i.e., SLM samples were processed in inert gas athmosphere (argon) couple with high scanning speeds and high thermal gradients leading to fast cooling rates; while EBM was done in vacuum where the powder bed and the surrounding powder particles are heated up to 700 °C, thus lowering the temperature gradients, hence, a lower cooling rate than SLM [24]. It is well known that α′ martensite is main strengthening phase in Ti6Al4V titanium alloys, but is less ductile than acicular α phase [35,36]. Therefore, following printing, SLM samples are often heat treated to improve its ductility [1,21]. The wrought samples, showed traces of rolling effect as indicated by relatively elongated (pancake-shaped) grains on the longitudinal plane sample. The structure contained both α′ martensite and acicular α. The difference in the microstructures, coupled with anisotropy associated the build direction [6,37,38,39] certainly play a role on the hardness variations as reported below.

SLM hardness ranged from 372 to 391 HV (Table 2 and Figure 6). Thijs et al. [20] reported hardness values between 350 and 520 HV, and suggested the high hardness values were obtained at slower scanning speed. For our exeriments, the samples were produced at a scanning speed of 1200 mm/s, while in Thijs et al. experiments, the scan speeds were between 50 and 200 mm/s [20].

For EBM, our samples showed slightly lower hardness compared with those of SLM, i.e., 319–360 HV (Table 2 and Figure 6). This hardness range is comparable to those reported elsewhere. Murr et al. [11] reported hardness around 360-410 HV, Chern et al. [15] showed a range of 304–388 HV, Karlsson et al. [17] 300–420 HV; and Galarraga et al. [18], and Hrabe and Quinn [33] reported 360–380 HV, depending on the built direction and the parameters used. Hardness of the wrought samples were between the those of EBM and SLM, and that explained the mixture of the presence of α′ martensite and acicular α on the sample.

### 3.3. Tensile Tests and Fractography

In general, for the AM samples, SLM showed a slight higher strength in all built directions than those of EBM but both had comparable ductility. A summary by Lewandowski and Seifi [29] strongly suggested that samples produced by SLM are stronger than those of EBM which is in agreement with our results. Tensile strengths are mainly in the range of 800–1000 MPa and 900–1200 MPa for EBM and SLM, respectively [5], although He et al. [40] claimed to have achieved a tensile strength of SLM up to 1261 MPa. 

Figure 7 presents the stress-strain curves of the samples from SLM, EBM and wrought alloys in all three directions/build orientations, i.e., 0°, 45° and 90° where the effect on strengths is fairly obvious. In the case of SLM, the built direction affect both σ_y_ and σ_u_ where 45° direction being the strongest and transverse (0°) being the weakest and least ductile. This is in agreement with what was reported by Ren et al. [13] and Agius et al. [41]. It is believed that that the residual stress and the pore distribution was responsible for this (note that following hot isostatic pressing—HIP, the difference was eliminated [9,13]). For EBM materials, again the 45° showed higher strengths compared with 0° and 90° directions. Although it may not be significant but the elongation was lower in 0° compared with other directions for both AM-produced materials. However, a different effect of direction was observed on the wrought materials. The strengths were the highest on the 90° followed by 0° and 45°. This is understandable as the direction samples were pulled normal to the grain orientation. A study by Vilaro et al. [21] showed similar finding on the SLM-built samples, where transverse direction showed a slight higher strength (around 4%) compared to the longitudinal direction. The elongation of wrought samples were nearly double the elongation of both SLM and/or EBM with the longitudinal (0°) direction showed highest ductility than all other samples (Figure 7 and Table 3). Such variations in the mechanical properties on both the AM-built specimens and the wrought alloys was arguably associated with the anisotropy of the samples as a result from printing/scanning or from rolling directions for AM and wrought alloys, respectively [6,37,38,39]. It should also be born in mind that the AM samples had significantly higher surface roughness than wrought alloys, so the comparative tensile results combined the effects of bulk material and surface differences. 

With regards to the stiffness, the slope in the elastic region on the stress vs. displacement graph showed that SLM had a slightly higher stiffness than both EBM and wrought samples in all three orientations (Figure 7).

The results from tensile testing represented by the digital image corelation (DIC) analysis showed a clear indication that wrought materials had more ductility compared with AM materials (Figure 7, Figure 8, Figure 9 and Figure 10). EBM samples had elongation of up to 7%, while SLM samples reached up to 10% and wrought samples experienced elongation up to 15% before fracture. The relatively lower elongation on EBM samples could be related to the large grains compared with those of SLM and wrought alloys. While the higher elongation particularly observed on the 90° of SLM samples is, arguably, due to the favourable build orientation (Figure 1) coupled with smaller grains (Figure 3, Figure 4 and Figure 5). The wrought alloys showed highest elongation in all directions due to their uniform grain size and more dense (no pores) as it has gone through rolling process. 

From the DIC images it is also obvious that the highest strain took place in the reduced sections area, hence, fracture locations. The local strain where fracture took place are indicated in red, and it showed somewhat higher than the total elongation. While the AM samples showed negligible necking (Figure 8 and Figure 9), the wrought samples (Figure 10) clearly showed noticeable necking mechanism prior to fracture, i.e., up to 54% compared to 19.5% and 14% maximum for SLM and EBM samples, respectively. 

Macrographs of the fracture surfaces from tensile test samples are shown in Figure 11, Figure 12 and Figure 13. Both SLM and EBM samples showed either flat or 45° angle with fairly little (or no) evidence of necking, while the wrought samples showed significant necking prior to fracture where a clear cup and cone shape can be seen on Figure 13.

The fracture surfaces of all samples showed dimples (Figure 14) resulting from micro-void coalescence indicating some ductile behavior. Similar features were also reported by others, for example in [25,26]. 

Dimples on the wrought samples were more regular and their distribution was also more uniform. The SLM samples had some small spherical gas pores, while the EBM samples showed some relatively larger irregular (crack-like) pores. From the shape of the pores, it can be assumed that the cause of those pores were different, e.g., gas entrapment around the melt pool on the former [4], and incomplete melting and solidification due to insufficient energy input on the latter [4]. The difference shapes and size of pores in EBM and SLM could be associated with the scanning strategy as well as the size of the powder and /or the parameters used which may produce different energy density. Gong et al. [26] reported the effect of processing parameters (led to different energy density) which caused different size and type of defects. Various types of AM defects have been reported such as balling [4], gas porosity and lack of fusion [42,43]. Shipley et al. further explained that SLM is particularly prone melt pool instability as well as wrong parameters, it may lead to various defects such as balling and porosity (spherical or sharp) being the two main ones [4]. As for the gas pore, it is likely due to gas entrapment in the melt pool [18]. The lack of fusion, particularly those of sharp-tip, may act as local stress concentration where cracks could be initiated which lead to a reduced strength [9,42,43] as well as fatigue properties [9,22,44,45,46,47].

## 4. Summary

A comparison of Ti6Al4V manufactured by SLM and EBM with wrought alloy has been investigated. Both the SLM and EBM were in the as-printed condition, while the wrought alloy was in the as-received condition. The summary from this investigation are as follows:The microstructures of the samples were significantly different with α’ martensite on the SLM, acicular α on EBM and combination of both on the wrought alloy.EBM samples had higher surface roughness (Ra) compared with both SLM and wrought alloy. For AM materials, the surface roughness was associated with the size of the particle and the improper melting.SLM samples were comparatively harder than wrought alloy and EBM.Tensile strength of the wrought alloy was higher in all directions except for 45°, where the SLM samples showed higher strength than both EBM and wrought alloy for the same direction. The ductility of the wrought alloy was consistently higher than both SLM and EBM indicated by clear necking feature on the wrought alloy samples. The variations in mechanical properties is believed to be associated with anisotropy on the samples.All the fracture surfaces showed some dimples indicating with wrought samples had more uniform dimples than the AM samples. Defects, i.e., porosity and crack-like defects were present on both SLM and EBM samples.

## Figures and Tables

**Figure 1 materials-14-03603-f001:**
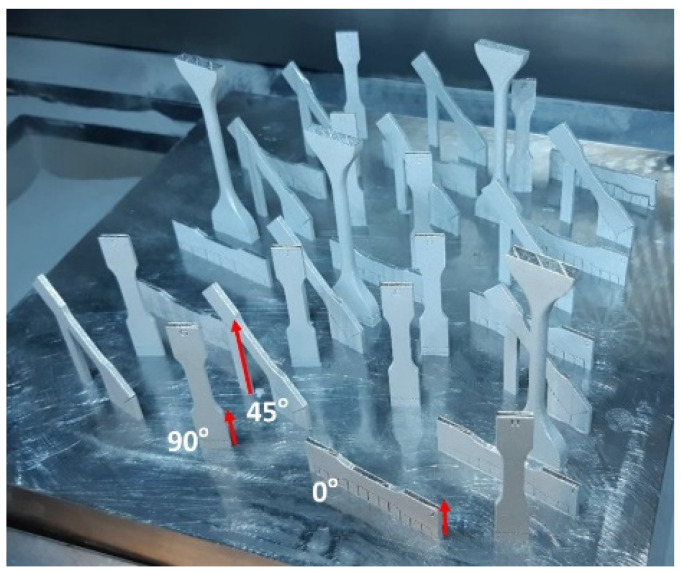
Diagram showing samples after printing by SLM (similar printing diagram was reported for EBM). Note: build orientation is shown by the red arrows.

**Figure 2 materials-14-03603-f002:**
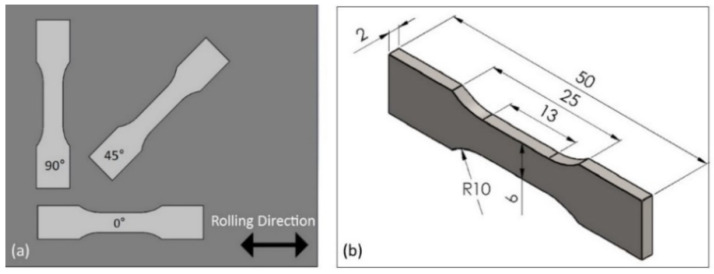
(**a**) Dog-bone sample orientation for the wrought alloy and (**b**) dimensions for all samples (in mm).

**Figure 3 materials-14-03603-f003:**
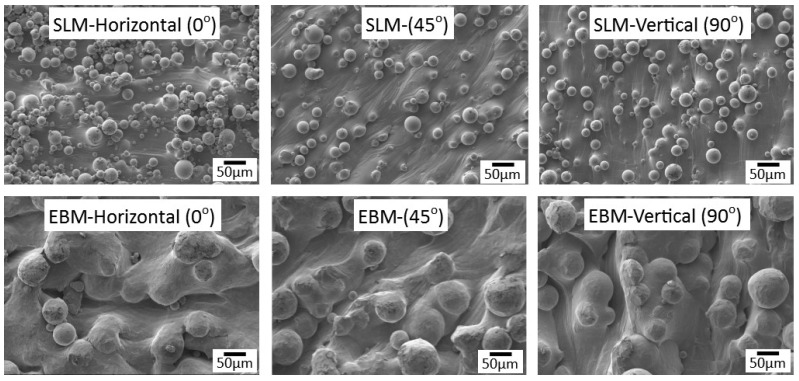
Surface morphology of samples built by SLM and EBM in various direction.

**Figure 4 materials-14-03603-f004:**
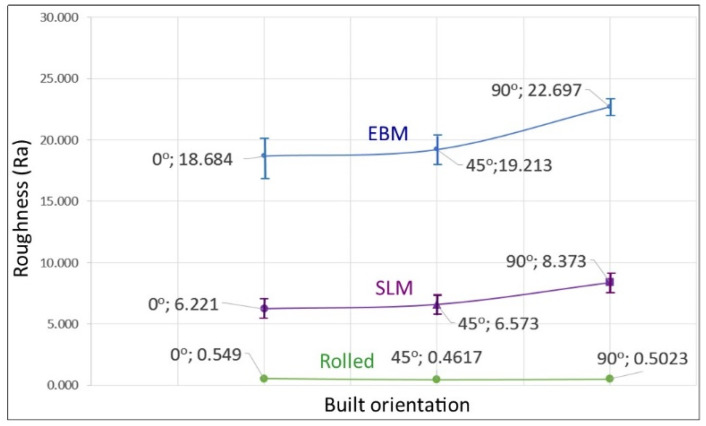
Surface roughness, Ra, values (in µm) of EBM and SLM samples built in different orientations. Note: the wrought sample was rolled in longitudinal direction and were cut in 0° (longitudinal/horizontal), 45° and 90° (transverse) directions.

**Figure 5 materials-14-03603-f005:**
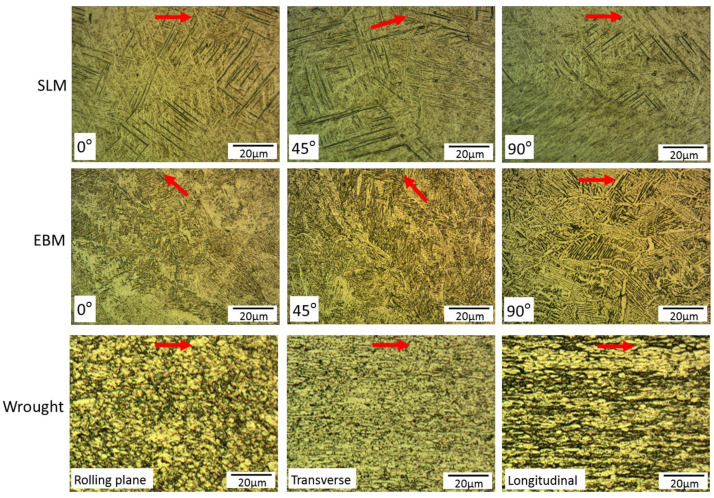
Optical micrographs of SLM, EBM and wrought samples. Note: build orientation for SLM and EBM (and rolling direction—wrought alloys) are indicated by the red arrows.

**Figure 6 materials-14-03603-f006:**
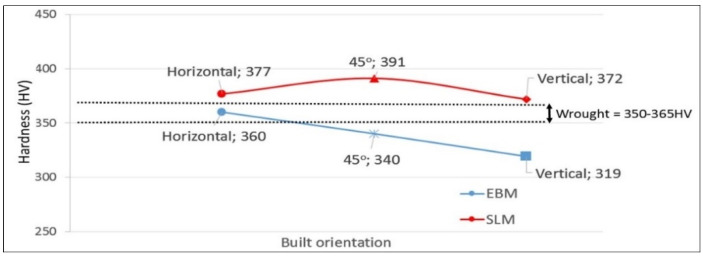
Hardness values (HV_300_) of EBM and SLM samples built in different orientations. Hardness values of wrought samples are indicated.

**Figure 7 materials-14-03603-f007:**
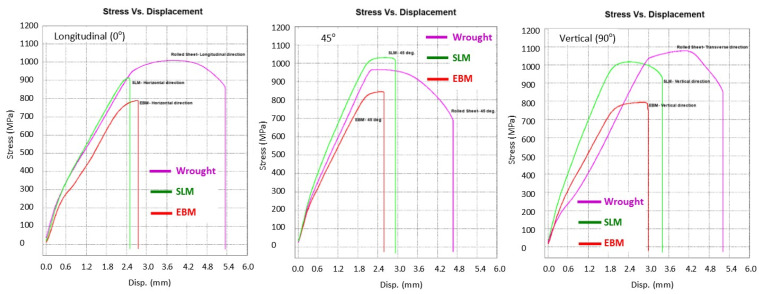
Stress vs. displacement diagrams of samples in 0°, 45° and 90° directions.

**Figure 8 materials-14-03603-f008:**
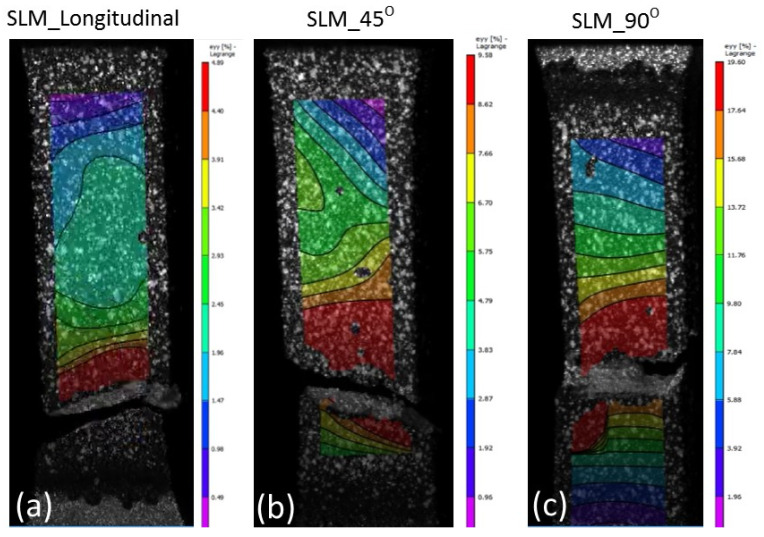
DIC images showing all samples at fracture states of SLM material. Note the maximum elongations at fracture locations were about (**a**) 5%, (**b**) 10% and (**c**) 19.5% on the 0°, 45° and 90°, respectively.

**Figure 9 materials-14-03603-f009:**
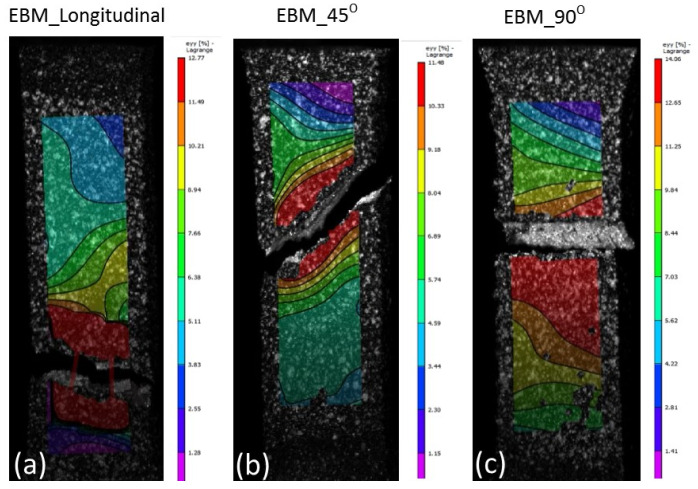
DIC images showing all samples at fracture states of EBM material. Note the maximum elongations at fracture locations were about (**a**) 13%, (**b**) 12% and (**c**) 14% on the 0°, 45° and 90°, respectively.

**Figure 10 materials-14-03603-f010:**
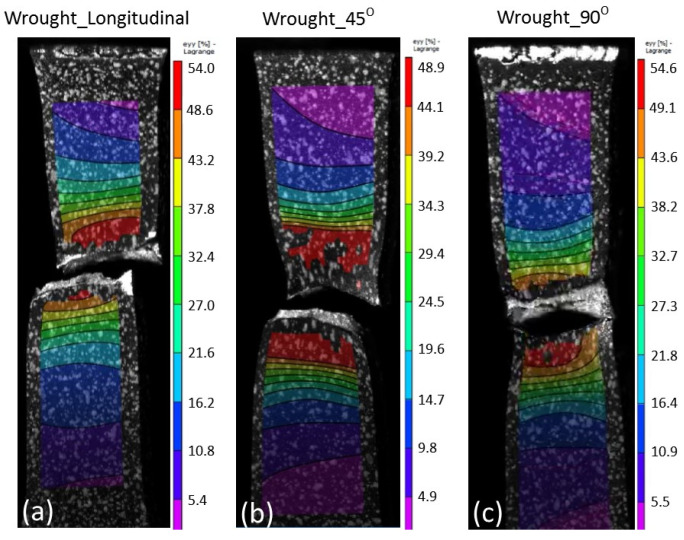
DIC images showing all samples at fracture states of wrought material. Note the maximum elongations at fracture locations were about (**a**) 54%, (**b**) 50% and (**c**) 54.5% on the 0°, 45° and 90°, respectively.

**Figure 11 materials-14-03603-f011:**
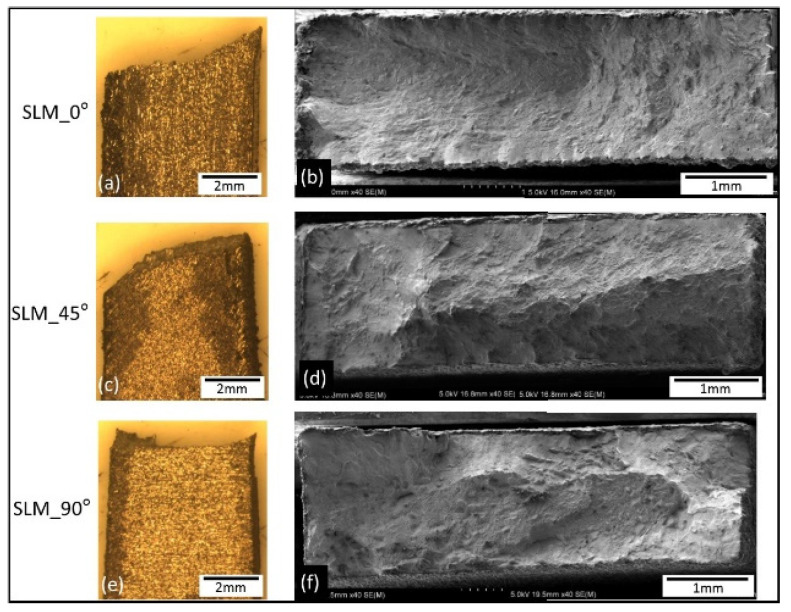
Macrographs showing side view of the tensile tested samples (**a**,**c**,**e**) and low magnification images of the fracture surfaces of the SLM samples (**b**,**d**,**f**).

**Figure 12 materials-14-03603-f012:**
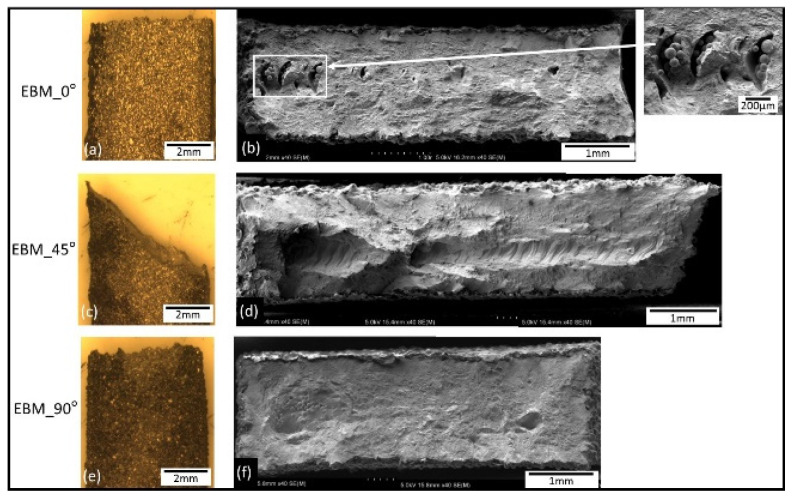
Macrographs showing side view of the tensile tested samples (**a**,**c**,**e**) and low magnification images of the fracture surfaces of the EBM samples (**b**,**d**,**f**). Note the un-melted particles on EBM_0° also reported in [16,20,27].

**Figure 13 materials-14-03603-f013:**
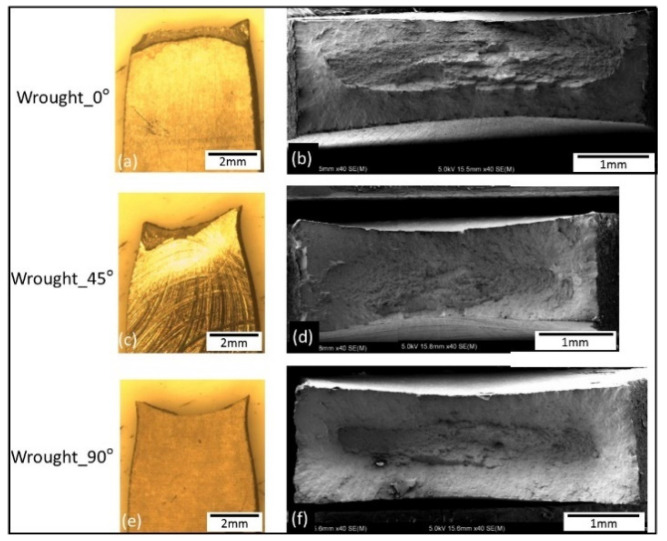
Macrographs showing side view of the tensile tested samples (**a**,**c**,**e**) and low magnification images of the fracture surfaces of the wrought alloy (**b**,**d**,**f**).

**Figure 14 materials-14-03603-f014:**
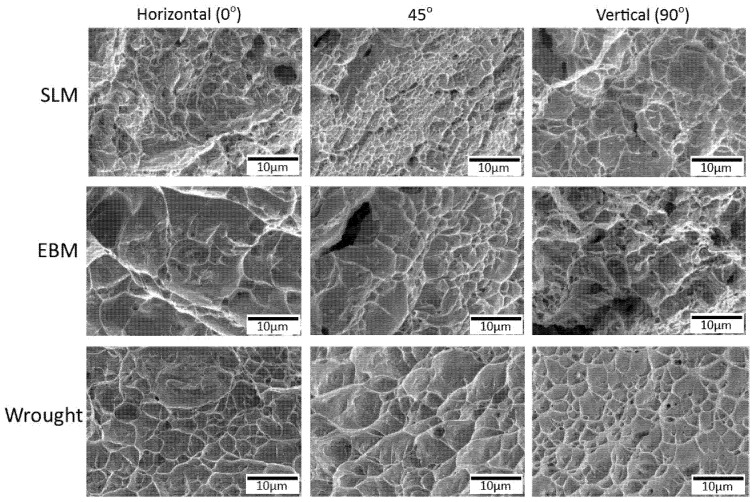
SEM micrographs showing high magnification images of fracture surfaces from SLM, EBM and wrought alloy samples. Note the presence of pores on the SLM and EBM samples.

**Table 1 materials-14-03603-t001:** Chemical composition of the titanium alloys (wt.%).

	Ti	Al	V	Mo	Cr	Fe	C	O	N
Ti64-SLM	Bal	6.27	4.15	<0.01	0.02	0.21	0.02	0.12	<0.02
Ti64-EBM	Bal	5.97	4.25	<0.01	0.02	0.16	<0.02	0.09	<0.02
Ti64-wrought	Bal	6.08	3.98	<0.01	0.02	0.17	0.02	0.05	<0.01

**Table 2 materials-14-03603-t002:** Average hardness (HV300) for AM samples *.

Samples	0°	45°	90°
SLM.	377	391	372
EBM	360	340	319

* Hardness for wrought samples ranged from 350 to 365 HV regardless of the orientatin.

**Table 3 materials-14-03603-t003:** Tensile testing results (three samples were tested for each material in each direction).

Samples/Properties	Yield Strength (MPa)			Tensile Strength (MPa)			Total Elongation (%)			Elongation at Fracture Location (%)		
	0°	45°	90°	0°	45°	90°	0°	45°	90°	0°	45°	90°
SLM	898	990	960	950	1036	1009	4	8	10	5	10	20
EBM	770	810	760	811	847	800	5	6	7	13	12	15
Wrought (as received)	965	814	1021	1016	965	1074	15	12	15	54	50	55
Wrought (STA) *	1052	970	990	1040	1008	1075	10	9	10	N/A	N/A	N/A

* STA = 960 °C/15 mins, WQ + 560 °C/4 h, AC.

## Data Availability

Not applicable.

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
