# Peer review of "Directionally-Dependent Mechanical Properties of Ti6Al4V Manufactured by Electron Beam Melting (EBM) and Selective Laser Melting (SLM)"

_materials, 2021, doi:10.3390/ma14133603_

Round 1
Reviewer 1 Report
The manuscript "Directionally-dependent Mechanical Properties of Ti6Al4V Manufactured by Electron Beam Melting (EBM) and Selective Laser Melting (SLM)" presents the comparative analysis of the mechanical properties and fracture behavior of the Ti64 alloy. In the current study the samples of the mentioned alloy were produced by the EBM and SLM techniques. Each of these methods was used to manufacture samples with three different production direction. Samples, obtained by additive manufacturing techniques, were then compared with the samples of the wrought alloy of the same chemical composition. Conclusions about preferable scheme of the samples manufacturing are drawn.
Manuscript is well-structured and strongly written except for a few cases of missed words and minor syntax errors. Reviewer would like to address to the authors few notes:
- Reviewer did not observe melt pools on the structure images. What could be the reason for that?
- Authors mention in the Experiments section that the difference in the surface roughness might have an impact on the tensile tests behavior of the samples. In that case, why not polish samples before tensile testing? That will eliminate surface effect and thus tensile tests will actually show the difference in the microstructure within the materials.
- Following the previous question: the sample, produced by SLM, is not homogeneous in its structure. Along the wall of the material the structure of the sample is different. In case of this experiment, the shape and the structure of this near-to-surface layer was different for the samples, produced with 0°, 45° and 90°. This difference will impact the tensile tests. How do authors address that issue?
- Did authors consider printing the bulk samples and only after that cutting the dog bone tensile tests samples out of them?
- On Figure 7 the left green curve seems to be obtained from the brittle fracture. Are these curves the average ones, or only one sample for each curve was tested?
Apart from that, this work is solidly done. The reviewer would recommend it for the publishing.
Author Response
We thank the reviewer for the feedback. We have added and modified as per suggestion, and have also added another reference. Please see the response - attached

Reviewer 2 Report
The article presents an topic about the Mechanical Properties of Ti6Al4V Manufactured by Electron Beam Melting (EBM) and Selective Laser Melting (SLM), however there are points that need to be clarified and summarized.
- Abstract: I suggest improving the abstract, making clear the purpose and the main results that have been achieved with this study.
- Introduction: The introduction is very general. It should be improved and clearly explain the purpose and final objective of this study with appropriate references. In a research paper, it is expected that introduction section briefly explains the starting background and, even more important, the originality (novelty) and relevancy of the study is well established. Once this is done, hypothesis and objectives of the study need to be addressed, as well as a brief justification of the conducted methodology.
- Results and discussions (Section 3): Put in this section only the part of the results obtained in this research for in another separate section (discussion), compare and discuss those results explaining the main advances obtained by comparing and discussing them with the results obtained by the studies of other authors..
- Discussion Section: Create a separate Discussion section. The Discussion section should compare the study by clearly comparing the results obtained by the authors with other studies conducted by other authors.
- Conclusions Section: Improve the conclusions section, it is very general and does not clearly explain the main objectives achieved in this research. The conclusions section should present in a clear and summarized way the main parts obtained with this study and the main contributions.
Author Response
We modified the abstract, introduction and conclusion - please see the attached document

Reviewer 3 Report
The manuscript will undoubtedly be of interest to readers. However, there are some points in the manuscript that need to be improved:
1) In introduction indicates that there are several works [22-28] that compare EBM and SLM processes. I think that the results of these works need to be described in more detail and correlated with the results obtained by the authors of the manuscript.
2) Materials and methods. Figure 2. How were the specified sample shapes made for all three discussed processes?
3) Materials and methods. The brand of the scanning electron microscope used for the fractographic analysis is not indicated.
4) Materials and methods. Three points for determining the microhardness of samples are not enough, especially for samples made by additive technologies (see also doi: 10.3390 / ma12193269)
5) Table 1. It is possible that it should be moved to the section "Materials and methods". For the manufacture of samples, were you used ready-made titanium alloy powders? Then it is necessary to indicate the manufacturer of metal powders. Why were powders of different fractions used for different methods of additive technologies?
6) Figure 3. It is possible that the different morphology of the samples is associated with different fractions of the powders used and is in no way connected with the methodology of the processes being carried out. This needs to be discussed in the text of the manuscript.
7) The numerical estimate of the porosity is not indicated. It is need to be done.
8) Conclusion. In the course of the work, quite interesting results were obtained, but unfortunately in the conclusion the answer to the question was not given: Why were different characteristics of the samples obtained at different orientations at 0°, 45° and 90°?
Author Response
We have modified the abstract, added the requested information, and have also added a few new references.

Reviewer 4 Report
The result is good but since this work is an experimental investigation only, thus more information and details about the experimental setup are needed.
A) About the 3D-DIC setup:
- How you speckle the specimen and what is the size of the spot in pixel?
- How you did the calibration?
- What are the step and subset values?
- What is the error range, and how you reduce it?
- How the cameras triggered?
- It is useful to add the table and real experimental figure to show all this information.
B) Why you used stress vs displacement? Please, use the force vs displacement or stress vs strain (from DIC). Then you can find the elastic modulus for each configuration, thus you can compare to the mechanical properties at the macroscale.
Author Response
Details on DIC have been explained - please see the attached document

Round 2
Reviewer 3 Report
To increase the scientific value of the manuscript, I would recommend adding some reasoning to the text of the manuscript:
1) Fig. 5. The structure of samples obtained using additive technologies depends on the build orientation. This also affects the anisotropy of the properties of the materials obtained. Reflect on this topic.
2) Fig. 14. Correlate the numerical values of porosity with fractographic analysis. It is also necessary to estimate the size of the cells on the resulting fractures.
Reviewer 4 Report
The revised version has been improved according to the comments. Hence, I recommend accepting this revised version in the materials.
